# Online Map Vectorization for Autonomous Driving: A Rasterization Perspective

**Gongjie Zhang**[†1]    **Jiahao Lin**[†1]    **Shuang Wu**[†1]    **Yilin Song**[1]    **Zhipeng Luo**[1,2]

**Yang Xue**[1]    **Shijian Lu**[2]    **Zuoguan Wang**[✉1]

[1]Black Sesame Technologies    [2]Nanyang Technological University, Singapore

† : equal contribution.    ✉ : corresponding author.    Project Page: https://github.com/ZhangGongjie/MapVR

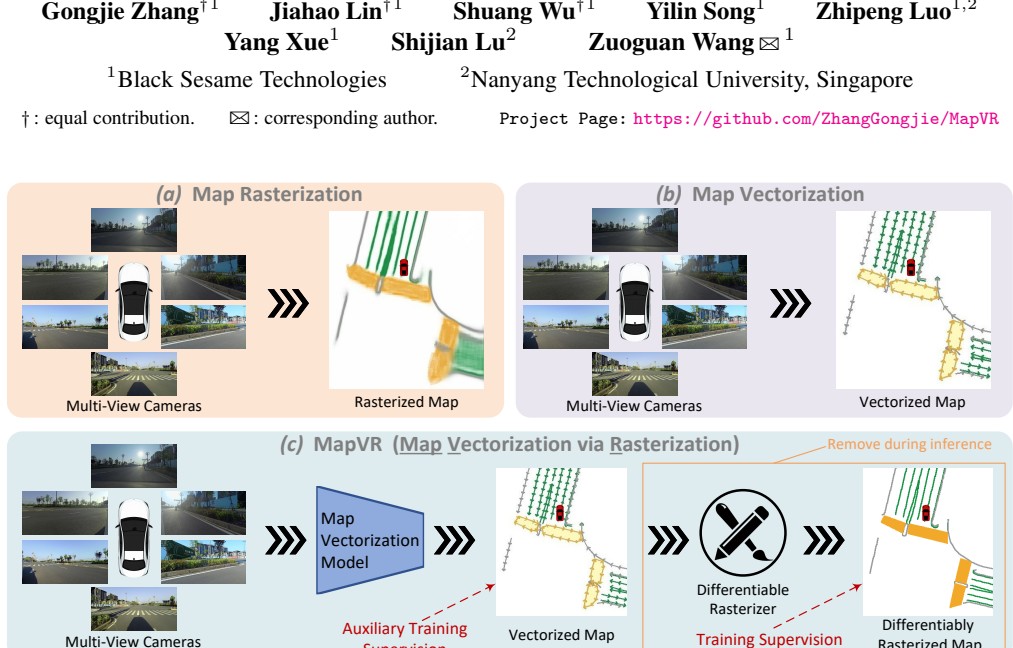

Figure 1: *(a)* Map rasterization produces HD semantic maps as output via semantic segmentation in bird's-eye view (BEV). *(b)* Map vectorization directly predicts compact and instance-level vectorized map elements that are better suited for autonomous driving systems. *(c)* MapVR employs differentiable rasterization to bridge vectorized and rasterized HD map representations, enabling more precise and accurate vectorized HD maps for reliable autonomous driving.

## Abstract

Vectorized high-definition (HD) map is essential for autonomous driving, providing detailed and precise environmental information for advanced perception and planning. However, current map vectorization methods often exhibit deviations, and the existing evaluation metric for map vectorization lacks sufficient sensitivity to detect these deviations. To address these limitations, we propose integrating the philosophy of rasterization into map vectorization. Specifically, we introduce a new rasterization-based evaluation metric, which has superior sensitivity and is better suited to real-world autonomous driving scenarios. Furthermore, we propose MapVR (Map Vectorization via Rasterization), a novel framework that applies differentiable rasterization to vectorized outputs and then performs precise and geometry-aware supervision on rasterized HD maps. Notably, MapVR designs tailored rasterization strategies for various geometric shapes, enabling effective adaptation to a wide range of map elements. Experiments show that incorporating rasterization into map vectorization greatly enhances performance with no extra computational cost during inference, leading to more accurate map perception and ultimately promoting safer autonomous driving.

37th Conference on Neural Information Processing Systems (NeurIPS 2023).

# 1 Introduction

Online high-definition (HD) map construction is essential for autonomous driving systems, as it supplies real-time and comprehensive information about the vehicle's surroundings, such as lanes, curbsides, and crosswalks. It serves as the foundation for the vehicle's navigation, planning, and decision-making processes, and is integral to the effective functioning of self-driving vehicles.

Existing online HD map construction methods fall into two classes: map rasterization and map vectorization. Map rasterization [37, 40, 12, 59, 19, 60, 36, 50] is straightforward: as shown in Fig. 1 (a), it models HD map construction as a semantic segmentation task in bird's-eye view (BEV), rasterizing the surroundings into semantic maps as output. However, rasterized maps are not ideal representations for autonomous driving, as they lack instance-level and structural information, and require extensive post-processing to be consumed by subsequent navigation and decision-making modules. To address these limitations, map vectorization (Fig. 1 (b)) emerges as a popular solution for constructing HD maps. HDMapNet [16] and SuperFusion [6] employ complex post-processing to group pixels from rasterized maps into vectors. The recent VectorMapNet [26] and MapTR [20] directly predict map elements as vectorized point sets, achieving better accuracy with faster runtime.

Both VectorMapNet [26] and MapTR [20] utilize a sparse point set representation, where each map element is parameterized as a fixed-length vector of equidistantly sampled points, with L1 loss applied to supervise regression predictions. While this approach is simple and intuitive, we empirically observe that it is often suboptimal due to several reasons. *First*, as shown in Fig. 2, the sparse point set representation is often lacking in precision, particularly when dealing with sharp bends or complex details of map structures, resulting in significant parameterization errors.[1] *Second*, learning with equidistant points as regression targets causes ambiguous supervision, because the intermediate points often lack clear visual clues. *Third*, relying solely on the L1 loss for regression supervision causes the model to overlook fine-grained geometric variations, yielding overly smooth predictions that are insensitive to local deviations. Likewise, the current evaluation metric, which relies on Chamfer distance among point sets, tends to overlook minor deviations and geometric details. For autonomous driving, where precision is a matter of life and death, existing methods and metric for map vectorization are still inadequate.

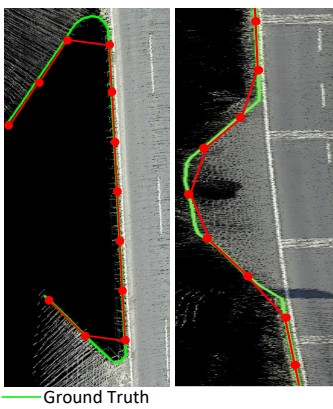

— Ground Truth
•—•— Equidistant-Point-Parameterized Line

Figure 2: Inaccurate map elements caused by the parameterization of sparse equidistant point sets.

To address these limitations, we reintroduce the philosophy of rasterization into map vectorization, to bring back the advantages of precision in HD map modeling while keeping the merits of vectorized outputs. We believe that rasterization can offer complementary benefits to map vectorization.

With the above motivation, we first design a new rasterization-based evaluation metric for map vectorization, which is more sensitive to minor deviations and better suited for practical driving scenarios. Unlike existing metric that uses Chamfer distance to determine if a map element matches the ground truth, we rasterize both the predicted and ground truth map elements into HD maps, and then use mean intersection-over-union (mIoU) to decide whether they match. This metric aligns better with human perception, takes into account the actual shape and geometry of individual map elements, and offers increased sensitivity to minor discrepancies.

We further present MapVR (Map Vectorization via Rasterization), a novel framework for precise HD map vectorization. MapVR can be integrated with any architecture that directly predicts vectorized map elements [16, 20]. Unlike existing map vectorization methods, our MapVR applies differentiable rasterization to vectorized output (ordered point sets) during training, transforms each vectorized map element into an HD map, and adds segmentation supervision on the rasterized HD maps. The proposed MapVR, sharing the philosophy with our aforementioned evaluation metric, enables more precise and detailed supervision, thus significantly boosting precision. It also provides more reasonable supervision, as it removes the ambiguity caused by equidistance. MapVR can also adapt

---

[1] While increasing the vector parameterization's dimensionality could resolve this issue theoretically, such an approach has been found to be not helpful in practice, as observed in MapTR [20].

to a wide range of map elements with specially designed geometry-aware differentiable rasterization strategies, showing strong scalability. At the inference stage, the additional differentiable rasterization can be simply removed, and the network's vectorized output can be employed as the final result. As our method does not introduce any additional computational overhead during inference, it maintains high efficiency, while delivering more accurate and robust map construction results.

The contributions of this work are summarized as follows:

- We propose a novel rasterization-based evaluation metric for map vectorization that exhibits increased sensitivity to minor deviations, providing a more accurate and reasonable assessment of map vectorization performance in real-world driving scenarios.
- We propose MapVR (Map Vectorization via Rasterization), a novel framework that seamlessly combines differentiable rasterization with existing map vectorization approaches. MapVR substantially improves the precision for map vectorization, demonstrates robust scalability for diverse map elements, and incurs no extra computational overhead during inference.
- The proposed MapVR framework and evaluation metric pave the way for future research and advancements in map vectorization for autonomous driving applications, demonstrating the complementary benefits of rasterization to map vectorization.

## 2 Related Work

**HD Map Construction.** Understanding the vehicle's surrounding environment, including lanes, curbsides, crosswalks, and road topology, plays a central role in the navigation and decision-making of autonomous driving. Such driving scene information is usually provided by high-definition (HD) maps. Conventionally, HD maps are constructed offline using SLAM-based methods [58, 32, 41, 42] with complex pipelines. Recently, with the emergence of the bird's-eye-view (BEV) perception [19, 27, 51, 36, 30, 48, 29], the focus has shifted towards online HD map construction, which generates maps around ego-vehicle from vehicle-mounted sensors (*e.g.*, cameras) on the fly.

Currently, there are two prevalent paradigms in online HD map construction: map rasterization and map vectorization. Rasterization methods [37, 40, 12, 59, 19, 60, 36, 50] generate HD maps via semantic segmentation in BEV, which have good sensitivity to details. However, the lack of vital instance-level information and lane topology limits the utility of rasterized maps in downstream tasks like navigation and planning. On the other hand, map vectorization addresses this limitation by producing vectorized map elements. HDMapNet [16] and SuperFusion [6] employ post-processing to group pixels from rasterized maps into vectorized elements. Moreover, VectorMapNet [26] proposes to directly predict map elements as vectorized point sets in an auto-regressive manner, achieving superior performance. And MapTR [20] – the current state of the art, further proposes a unified permutation-equivalent modeling approach to model the HD map elements, achieving superior accuracy. Furthermore, MapTR [20] achieves real-time efficiency with its one-stage and parallel framework. However, despite the recent progresses, vectorized maps still often exhibit minor deviations that can be critical in autonomous driving, where safety is of utmost importance.

**Lane Detection.** Lane detection, which can be seen as a sub-task of online HD map construction, concentrates on map elements like lanes and curbsides. Most existing lane detection research targets 2D camera views, and can be categorized into several modes, including pixel-level segmentation [33, 39, 35, 52], anchor-point-based regression [43, 18], and curve-prior fitting using polynomial [46, 44, 23] or Bezier curves [8]. With the recent progresses in BEV perception, some lane detection methods [10, 4, 1, 47, 13] have also extended into 3D, perceiving lanes in BEV, aligning more closely with online HD map construction. Nonetheless, lane detection remains somewhat limited, perceiving only highly-regularized line-shaped map elements. In contrast, map vectorization has better scalability and adaptability with fewer assumptions, making it a better fit for real-world autonomous driving.

**Differentiable Rasterization.** Rasterization, a concept from computer graphics, refers to the process of rendering vector graphics representations (point coordinates or math formulas) into raster images (a series of pixels) for display on computer screens [45]. Typically, rasterization is non-differentiable [38, 9]. Fortunately, recent advances in graphics and vision [28, 24, 5, 25, 21, 17, 34, 15] have achieved differentiable rasterization, bridging the gap between vector graphics and raster image through backpropagation. In this work, we make the first attempt to adapt differentiable rasterization to the map vectorization task to bridge vectorized outputs and rasterized HD maps. It enables more refined and comprehensive supervision and yields predictions with improved precision.

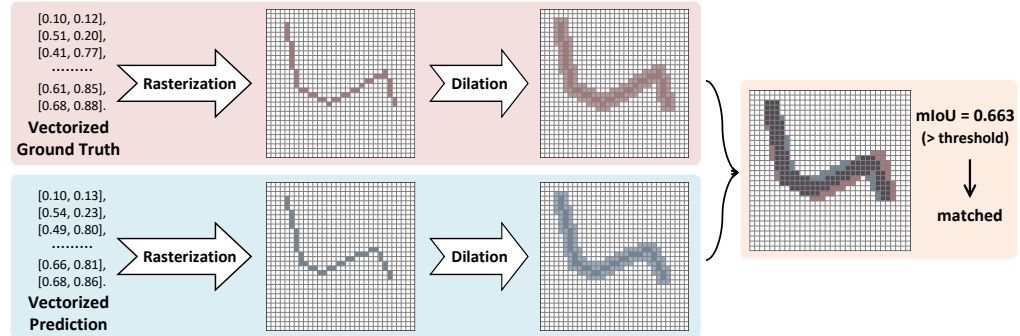

Figure 3: Illustration of our proposed rasterization-based approach for determining the match between ground truth and predicted vectorized map elements.

## 3 A Rasterization-Based Evaluation Metric for Map Vectorization

### 3.1 Review of Existing Chamfer-Distance-Based Evaluation Metric

Map vectorization requires instance-level evaluation, similar to object detection [7, 22, 53, 54, 3, 55, 57, 56, 61]. Thus, current map vectorization works [16, 6, 26, 20] adopt Average Precision (AP) to evaluate the map construction accuracy, using Chamfer distance to determine whether the predicted map element and the ground truth map element match.

Specifically, Chamfer distance $D_{\mathrm{Chamfer}}(\cdot, \cdot)$ is a measure of dissimilarity between two unordered point sets, which quantifies the average distance between each point in one set to the nearest point in the other set. It can be formulated as:

$$D_{\mathrm{Chamfer}}(P, Q) = 0.5 \times (\frac{1}{|P|} \sum_{p \in P} \min_{q \in Q} |p - q|_2 + \frac{1}{|Q|} \sum_{q \in Q} \min_{p \in P} |p - q|_2), \tag{1}$$

where $P$ and $Q$ are the sets of points representing the predicted map element and the ground truth map element, respectively, $|P|$ and $|Q|$ are the cardinalities of point sets $P$ and $Q$, and $|p - q|_2$ denotes the Euclidean distance between points $p$ and $q$.

Despite its simplicity and ability to provide fair evaluation results, the following limitations of this metric make it inadequate for highly demanding scenarios such as autonomous driving: *1)* It is not scale-invariant; for smaller map elements such as stoplines, Chamfer distance error is consistently small, failing to provide a meaningful assessment. *2)* Chamfer distance solely relies on unordered point set distance, completely overlooking the shape and geometrical details of the map elements, thus yielding unreasonable results for many practical scenes, as shown in Fig. 4. These drawbacks call for the development of a more robust and accurate evaluation metric tailored to the stringent requirements of autonomous driving map vectorization.

### 3.2 Proposed Rasterization-Based Evaluation Metric

To address the aforementioned limitations, we introduce a rasterization-based evaluation metric that is more sensitive to minor deviations and better suited for real-world driving scenarios. While we still employ AP as our measurement, we adopt rasterization to precisely determine the matching between predicted and ground truth map elements.

As shown in Fig. 3, we demonstrate our metric using line-shaped map elements (*e.g.*, lanes and curbsides). First, both ground truth and predicted elements are rasterized into a polyline in HD maps. In our setup, considering the perception range of $\pm30$m on the y-axis and $\pm15$m on the x-axis, we set the spatial size of the HD map as $480 \times 240$, such that each pixel represents $0.125$m, satisfying the high-precision requirement of autonomous driving. To better accommodate inaccuracies in predictions with thin and elongated geometry, we then dilate the rasterized polylines by 2 pixels on each side, thereby introducing an appropriate degree of tolerance. Finally, to determine whether the ground truth and predicted map elements match, we calculate the intersection-over-union (IoU) of their respective rasterized HD representations. Similar to MS-COCO's metric [22], AP is calculated at multiple IoU thresholds. For line-shaped elements, we set the thresholds as $0.25 : 0.50 : 0.05$.

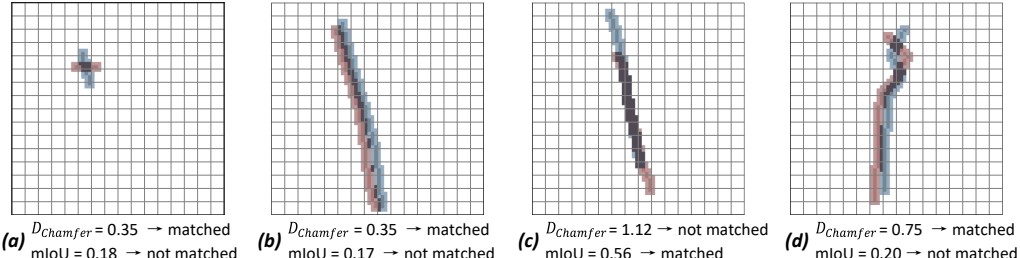

| | |
|---|---|
| *(a)* $D_{Chamfer}$ = 0.35 → matched | *(b)* $D_{Chamfer}$ = 0.35 → matched |
| mIoU = 0.18 → not matched | mIoU = 0.17 → not matched |
| *(c)* $D_{Chamfer}$ = 1.12 → not matched | *(d)* $D_{Chamfer}$ = 0.75 → matched |
| mIoU = 0.56 → matched | mIoU = 0.20 → not matched |

Figure 4: Evaluation quality comparison between the Chamfer-distance-based metric and our proposed rasterization-based metric on a few practical cases. Our metric is able to produce more reasonable evaluation suitable for autonomous driving applications.

It is worth noting that HD maps often contain elements other than lines, such as crosswalks, intersections, and carparks. These elements can be abstracted into polygons. To conduct an appropriate evaluation for polygon-shaped map elements, we apply specially-designed polygon-shaped rasterization instead of line-shaped rasterization, and compute AP over $0.50 : 0.75 : 0.05$.

### 3.3 Comparative Analysis and Discussion

**Evaluation Quality.** We examine the evaluation quality of the two metrics with a few practical examples. Fig. 4(a) displays a case involving a short stopline, where the prediction is perpendicular to the ground truth. The Chamfer distance metric judges a match, as it lacks scale-invariance. While the rasterization-based metric successfully recognizes the discrepancy based on their low IoU. Fig. 4(b) presents a scenario in which the predicted lane/curbside exhibits a minor horizontal deviation from the ground truth. Such deviations, even if small, pose critical dangers in real driving scenes. The Chamfer-distance-based metric considers the prediction as matched solely based on the small point-set distance. Conversely, our metric takes geometry into consideration, determining that they do not match. Fig. 4(c) illustrates a case with a vertical deviation between the prediction and ground truth, typically arising from occlusion. This situation is generally non-critical, as the map updates continuously as the vehicle moves forward. By incorporating shape and geometry knowledge, the rasterization-based metric evaluates more reasonably. Fig. 4(d) also verifies that our metric is more sensitive to small but critical errors. Collectively, these examples show that the rasterization-based metric offers superior sensitivity and is better aligned with practical autonomous driving scenarios.

**Computational Complexity.** The rasterization-based metric requires additional computation for rasterization but still runs acceptably fast. Empirically, the evaluation process on nuScenes Map [2] validation set takes $\sim 3$ minutes on our server equipped with an Intel Xeon Gold 6226R CPU.

## 4 MapVR (Map Vectorization via Rasterization)

### 4.1 Framework Overview

As shown in Fig. 1(c), MapVR is a novel and generic learning framework for map vectorization, which combines rasterization to leverage the fine-grained supervisory signal from the rasterized HD maps while retaining the benefits of vectorized representation. MapVR is parameter-free and thus can be easily integrated with various network architectures for map vectorization (*e.g.*, MapTR [20]).

Fig. 5 illustrates the overall framework of MapVR. During training, the base map vectorization model first generates vectorized representation for each map element. Then, MapVR produces an HD map by rendering the vectorized element with a specially-designed differentiable rasterizer. Finally, segmentation-based losses can be directly applied to the rendered HD maps, providing more granular supervision on the shape and geometry of the map elements, which leads to more precise results.

### 4.2 Differentiable Rasterization: Bridging Vectorized Representation and HD Semantic Maps

Rasterization serves as a vital bridge between vectorized representation and HD maps. Generally, rasterization is not differentiable due to the binary assignment that decides whether a pixel is covered by any shape primitive. Inspired by [24, 5, 17, 15], to enable fine-grained supervision signals directly from HD maps, we introduce a soft version of rasterization, which renders each vectorized map element into an HD mask while preserving the whole framework's differentiability.

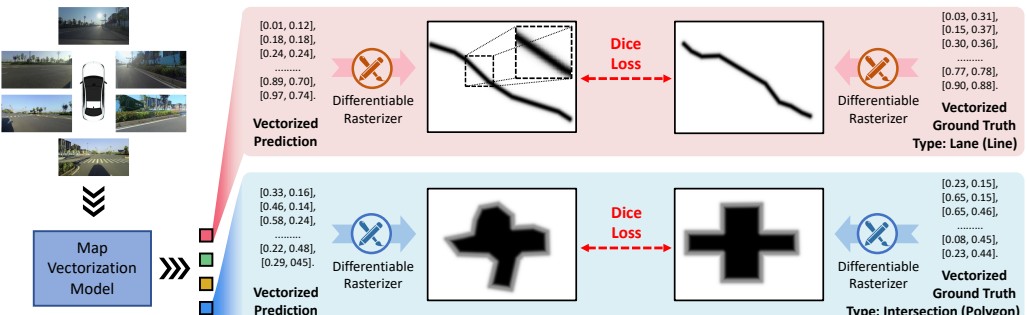

Figure 5: The learning pipeline of MapVR. MapVR utilizes a base model for vectorized map generation, followed by a customized differentiable rasterizer to produce HD maps, on which fine-grained, geometry-aware supervision is applied to enhance the precision of vectorized elements.

Concretely, for a line-shaped map element represented by an ordered point set $P$, we compute its softly-rendered mask $I_{\text{line}} \in [0,1]^{H \times W}$ with

$$I_{\text{line}}(x, y; P) = \exp\left(\frac{-D(x, y; P)}{\tau}\right), \tag{2}$$

where $D(x, y; P)$ denotes the closest distance from pixel $(x, y)$ to all segments of the polyline $P$, and the softness $\tau$ controls the rasterization smoothness. A larger $\tau$ yields smoother transitions between the polyline and empty regions, while a smaller $\tau$ leads to sharper, more distinct line boundaries.

While for polygon-shaped map elements like intersections, the rendered mask $I_{\text{polygon}}$ is computed as

$$I_{\text{polygon}}(x, y; P) = \sigma\left(\frac{C(x, y; P) \cdot D(x, y; P)}{\tau}\right), \tag{3}$$

where $D(x, y; P)$ is the closest distance from pixel $(x, y)$ to any boundary segment of the polygon $P$, and $C(x, y; P) \in \{-1, +1\}$ indicates whether pixel $(x, y)$ falls inside $(+1)$ or outside $(-1)$ the polygon. $\sigma(\cdot)$ denotes the sigmoid function. Similarly, the softness $\tau$ controls the transition smoothness of the rasterized values at the polygon boundary areas.

Our differentiable rasterizer (Eq. 2 & 3) transforms each vectorized map element into a rasterized HD mask representation in a parameter-free manner, which enables the learning of fine-grained shapes and geometric details through direct supervision on these rasterized HD masks.

### 4.3 Training and Inference Procedure

**Training.** Fig. 5 illustrates how differentiable rasterization is incorporated into the map vectorization framework. First, we use a base map vectorization model (*e.g.*, MapTR [20]) to predict a set of vectorized map elements. Then, instead of relying on L1 loss with equidistant points as targets as in [20, 26], we render both vectorized prediction and vectorized ground truth into rasterized HD masks, and apply supervision directly on the masks using dice loss [31]. Thanks to the differentiability of our designed rasterization processes (Eq. 2 & 3), the segmentation loss is able to guide the learning of vectorized predictions. Notably, this supervision is geometry-aware, as the rasterization procedure (line-shaped or polygon-shaped rasterization) is determined by the class of the target map element. The effectiveness of geometry-aware rendering is validated in Section 5.3. Moreover, the rasterization-based segmentation loss effectively weighs down the equidistance requirement (which is ill-posed due to the lack of clear visual clues), thus providing a more reasonable learning target.

In addition to the rendering-based loss, we include a direction regularization loss as an additional auxiliary loss. Specifically, we define the direction regularization loss on the vectorized output as

$$\mathcal{L}_{\text{dir}} = \sum_{i=1}^{N-2} \frac{<\overrightarrow{P_i P_{i+1}}, \overrightarrow{P_{i+1} P_{i+2}}>}{|\overrightarrow{P_i P_{i+1}}| \cdot |\overrightarrow{P_{i+1} P_{i+2}}|}, \tag{4}$$

where $P_i$ denotes the $i^{\text{th}}$ point in the predicted point set. It encourages the predictions to avoid unnecessary direction changes along adjacent segments. This effectively promotes a smoother point set to avoid back-and-forth patterns that are not penalized by the rendering loss, and also facilitates the allocation of more points in regions with higher curvature and fewer points in straight-line regions.

**Efficient Inference.** After training, the rasterization processes are no longer needed. Consequently, MapVR can enhance map vectorization without adding any extra computational cost during inference.

Table 1: Comparison of various map vectorization methods on nuScenes Map (basic) validation set.

| Method | Modality | Backbone | #Epochs | AP$_{\text{Chamfer}}$ | | | | AP$_{\text{raster}}$ | | | | FPS |
|---|---|---|---|---|---|---|---|---|---|---|---|---|
| | | | | ped | div | bdry | avg. | ped | div | bdry | avg. | |
| HDMapNet [16] | C | Effi-B0 | 30 | 14.4 | 21.7 | 33.0 | 23.0 | - | - | - | - | 0.8 |
| HDMapNet [16] | C & L | Effi-B0 | 30 | 16.3 | 29.6 | 46.7 | 31.0 | - | - | - | - | 0.5 |
| VectorMapNet [26] | C | Res-50 | 110 | 36.1 | 47.3 | 39.3 | 40.9 | 26.2 | 12.7 | 6.1 | 15.0 | 2.9 |
| VectorMapNet [26] | C & L | Res-50 | 110 | 37.6 | 50.5 | 47.5 | 45.2 | - | - | - | - | - |
| MapTR [20] | C | Res-50 | 24 | 46.3 | 51.5 | 53.1 | 50.3 | 32.4 | 23.5 | 17.1 | 24.3 | **18.4** |
| MapTR [20] | C | Res-50 | 110 | 56.2 | 59.8 | 60.1 | 58.7 | 43.6 | 35.6 | 25.8 | 35.0 | **18.4** |
| MapTR [20] | C & L | Res-50 | 24 | 56.4 | 61.8 | **70.1** | 62.7 | 46.4 | 39.2 | 50.0 | 45.2 | 7.2 |
| MapTR [20] + MapVR (Ours) | C | Res-50 | 24 | 47.7 | 54.4 | 51.4 | 51.2 | 37.5 | 33.1 | 23.0 | 31.2 | **18.4** |
| MapTR [20] + MapVR (Ours) | C | Res-50 | 110 | 55.0 | 61.8 | 59.4 | 58.8 | 46.0 | 39.7 | 29.9 | 38.5 | **18.4** |
| MapTR [20] + MapVR (Ours) | C & L | Res-50 | 24 | **60.4** | **62.7** | 67.2 | **63.5** | **52.4** | **46.4** | **54.4** | **51.1** | 7.2 |

- In modality, 'C' denotes multi-view camera input and 'C & L' denotes combined multi-view camera and LiDAR input.
- When LiDAR data is incorporated, PointPillars [14] serves as the backbone for processing the LiDAR data.
- The abbreviations 'ped', 'div', and 'bdry' correspond to the map elements of pedestrian crossing, divider, and boundary, respectively.

Table 2: Map vectorization performance on nuScenes Map (extended) validation set.

| Method | AP$_{\text{Chamfer}}$ | | | | | | | AP$_{\text{raster}}$ | | | | | | | FPS |
|---|---|---|---|---|---|---|---|---|---|---|---|---|---|---|---|
| | ped | stp | int | cap | div | bdry | avg. | ped | stp | int | cap | div | bdry | avg. | |
| MapTR [20] | 34.3 | 29.9 | 21.5 | 37.9 | 44.9 | 45.1 | 35.6 | 22.5 | 12.1 | 38.4 | 23.4 | 18.3 | 12.1 | 21.1 | **18.4** |
| MapTR [20] + MapVR (Ours) | **39.5** | **31.6** | **21.9** | **42.4** | **45.8** | **45.9** | **37.9** | **30.8** | **13.9** | **43.3** | **32.8** | **27.0** | **18.8** | **27.8** | **18.4** |

- All competing methods take multi-view cameras as input, use ResNet-50 [11] as the backbone, and are trained for 24 epochs.
- 'ped', 'stp', 'int', 'cap', 'div', and 'bdry' denote pedestrian crossing, stopline, intersection, carpark area, divider, and boundary, respectively.

## 5 Experiments

### 5.1 Experiment Setup

**Dataset and Evaluation Metrics.** MapVR is evaluated across multiple datasets, as outlined below.

1. nuScenes Map (basic) [2], which consists of two line-shaped map classes (lane divider and road boundary) and one polygon-shaped map class (pedestrian crossing). This dataset setup aligns with prior works on map vectorization [16, 6, 26, 20].

2. nuScenes Map (extended) [2], an extension of nuScenes Map (basic) that incorporates more complex map elements, such as intersection, stopline area, and carpark area.

3. Argoverse2 [49], a large-scale dataset featuring the same classes as nuScenes Map (basic).

4. 6V-mini-v0.4, our proprietary large-scale commercial dataset for autonomous driving, covering very complex driving scenes in real world. It includes three line-shaped classes (lane, curbside, and stopline) and two polygon-shaped classes (crosswalk and intersection).

Both Chamfer-distance-based metric (Section 3.1, denoted as AP$_{\text{Chamfer}}$) and the newly proposed rasterization-based metric (Section 3.2, denoted as AP$_{\text{raster}}$) are used for performance evaluation.

**Implementation Details.** All experiments, unless otherwise stated, are conducted with 8x NVIDIA RTX 3090 GPUs. As the proposed MapVR is a generic framework with no reliance on specific model architecture, we adopt MapTR [20], the state-of-the-art model for map vectorization, as the base model. Our implementation aligns with MapTR [20]. Please refer to the appendix for more details.

### 5.2 Experiment Results

**Results on nuScenes Map.** Table 1 compares MapVR with existing map vectorization techniques on nuScenes Map (basic). Even under the less sensitive AP$_{\text{Chamfer}}$ metric, our proposed MapVR delivers superior overall performance across various settings. The advantage of MapVR becomes even more pronounced under the more precise and autonomous-driving-oriented AP$_{\text{raster}}$ metric. Specifically, MapVR provides a notable 3.5% improvement over a fully-trained MapTR. When working with multi-modality inputs, MapVR obtains an even larger margin of 5.9%. As shown in Table 2, on the more challenging nuScenes Map (extended) dataset that includes more complex elements, our MapVR achieves superior performance across all map elements under both metrics. These results validate the substantial improvements brought by MapVR and its exceptional capability of adapting to challenging scenarios. It is noteworthy that these improvements are achieved without adding any additional computational burden during inference.

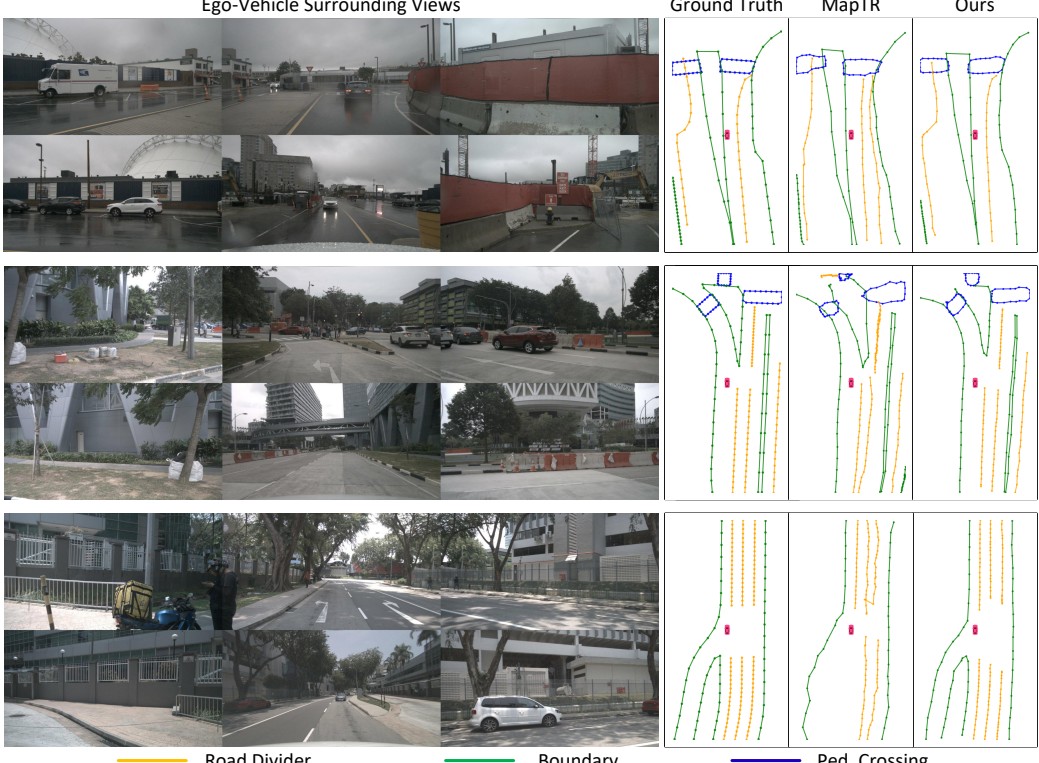

| Ego-Vehicle Surrounding Views | Ground Truth | MapTR | Ours |

Road Divider — Boundary — Ped. Crossing

Figure 6: Visualization of online HD map vectorization results. Our proposed MapVR demonstrates a superior ability in constructing more accurate maps, particularly for complex map elements and intricate details.

**Results on Argoverse 2.** Table 3 presents the performance comparison on the Argoverse2 dataset [49]. Note that in our setup, the height information for map elements is ignored. Our proposed MapVR method still achieves state-of-the-art performance, which verifies its robustness across multiple scenarios.

Table 3: Comparison of various map vectorization methods on Argoverse2 validation set.

| Method | $AP_{Chamfer}$ | | | | $AP_{raster}$ | | | |
| | ped | div | bdry | avg. | ped | div | bdry | avg. |
|---|---|---|---|---|---|---|---|---|
| HDMapNet [16] | 13.1 | 5.7 | 37.6 | 18.8 | - | - | - | - |
| VectorMapNet [26] | 38.3 | 36.1 | 39.2 | 37.9 | - | - | - | - |
| MapTR [20] | 54.7 | 58.1 | 56.7 | 56.5 | 22.1 | 32.6 | 24.0 | 26.2 |
| MapTR [20] + MapVR (Ours) | 54.6 | **60.0** | **58.0** | **57.5** | **23.5** | **36.5** | **30.2** | **30.1** |

• 'ped', 'div', and 'bdry' denote pedestrian crossing, divider, and boundary, respectively.

**Results on 6V-mini-v0.4.** Finally, we test MapVR on 6V-mini-v0.4, our proprietary commercial dataset that features highly intricate real-world driving scenes. As Table 4 shows, MapVR greatly enhances the performance on all map elements, which validates its efficacy and robustness in complex and real-world contexts.

Table 4: Map vectorization performance on 6V-mini-v0.4 dataset (our proprietary commercial dataset).

| Method | $AP_{raster}$ | | | | |
| | lane | curbside | stopline | crosswalk | intersection |
|---|---|---|---|---|---|
| MapTR [20] | 41.3 | 32.9 | 7.6 | 13.3 | 43.6 |
| MapTR [20] + MapVR (Ours) | **50.8** | **37.3** | **11.8** | **14.3** | **44.0** |

**Visualizations.** Fig. 6 visualizes the results of HD map vectorization and compares our method with MapTR [20]. For a fair comparison, both methods use the ResNet-50 [11] backbone and solely rely on multi-view camera images as input. It can be observed that our method yields more accurate HD maps, particularly in capturing intricate details and accurately representing complex or curved map elements. Conversely, while MapTR [20] produces generally correct vectorized maps, it inevitably exhibits deviations in finer details and struggles to precisely construct complex map elements. These observations reaffirm our motivation to incorporate the precise supervision from HD rasterization into map vectorization, which compensates for the inherent limitations caused by the sparse, equidistant point sets, thereby enhancing the precision of map vectorization.

Table 5: MapVR's ablation experiments on nuScenes Map (basic) validation set. All models employ ResNet-50 as backbones and are trained for 24 epochs. MapVR's default setups are marked in gray.

(a) Rasterization resolution. '×' denotes no rasterization.

| resolution | × | 64x32 | 128x64 | 180x90 | 256x128 | 320x160 |
|---|---|---|---|---|---|---|
| $mAP_{raster}$ | 24.3 | 21.5 | 29.8 | 30.4 | **31.2** | 30.9 |
| $mAP_{Chamfer}$ | 50.3 | 45.1 | / | 50.6 | **51.2** | 50.9 |

(b) Line rasterization softness $\tau$.

| line softness $\tau$ | 0.5 | 1.0 | 2.0 | 4.0 | 6.0 |
|---|---|---|---|---|---|
| $mAP_{raster(divider)}$ | 29.2 | 31.7 | **33.1** | 32.5 | 31.4 |
| $mAP_{Chamfer(divider)}$ | 48.0 | 50.3 | **54.4** | 53.3 | 52.8 |

(c) Regularization on point direction.

| regularization | None | w/ GT | w/ self |
|---|---|---|---|
| $mAP_{raster}$ | 29.5 | 29.3 | **31.2** |
| $mAP_{Chamfer}$ | 48.5 | 48.5 | **51.2** |

(d) Rasterization geometry-awareness.

| | all as lines | lines & polygons |
|---|---|---|
| $mAP_{raster(ped\ xing)}$ | 21.8 | **37.5** |
| $mAP_{Chamfer(ped\ xing)}$ | 34.9 | **47.7** |

(e) MapVR *vs.* parallel segm.

| | MapVR | parallel segm |
|---|---|---|
| $mAP_{raster}$ | **31.2** | 26.7 |
| $mAP_{Chamfer}$ | **51.2** | 48.1 |

• $mAP_{Chamfer}$ are added upon reviewers' kind suggestions. Entries marked with '/' are unavailable due to accidentally deleted checkpoints.

## 5.3 Ablation Study

**Rasterization & Rasterization Resolution.** As shown in Table 5a, incorporating rasterization enhances performance, following a general trend where higher resolutions yield better results. However, an exception is observed at the 64x32 resolution, which degrades the performance due to the lack of precise rasterization supervision at such a coarse resolution. Fig. 7 further presents the Precision-Recall curves, showing that MapVR leads to consistent performance gain under both metrics and, notably, a smaller gap under the two metrics. Conversely, the baseline exhibits a large drop under the stricter $AP_{raster}$. This proves the necessity of incorporating fine-grained supervision from rasterization.

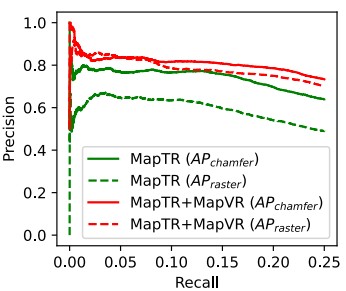

Figure 7: Comparison of P-R curves. MapVR narrows the performance gap under the coarse and strict metrics.

**Rasterization Softness $\tau$.** $\tau$ is a tricky hyper-parameter. It needs to be large enough to provide sufficient supervisory gradient while being small enough to ensure precise supervision. Empirically, polygon-shaped map elements are robust against various $\tau$, while line-shaped elements are not, due to their thin and elongated shapes. Table 5b studies the effect of different $\tau$ for line, taking the 'divider' class (a line-shaped map element) as an example.

**Auxiliary Regularization on Point Direction.** Table 5c studies the effect of the direction regularization loss described in Eq. 4, and also compares it with MapTR's directional loss [20], which uses the directions of ground truth equidistant points as targets. Results show that our direction regularization loss (w/ self) improves performance, proving its effectiveness in regularizing vectorized smooth point sets and allocating more points on higher curvature areas to improve precision. Conversely, the performance of our MapVR slightly degrades when using MapTR's regularization (w/ GT). This is because our MapVR's supervision from rasterization no longer requires the vectorized outputs to be equidistant.

**Geometry-Aware Rasterization.** Table 5d shows that simply rendering all map elements into lines severely impairs performance. The performance drop mainly comes from polygon-shaped elements (ped_crossing: 37.5%→21.8%). This verifies the necessity of geometry awareness in rasterization.

**Why Not Introduce HD Supervisory Signals from an Auxiliary Segmentation Task?** A simple alternative to incorporate fine-grained supervision from rasterization is to append an additional parallel segmentation branch as an auxiliary task (dubbed as 'parallel segm'). This has been verified effective in many works [8, 4, 1]. Table 5e compares MapVR with this strategy. While 'parallel segm' improves baseline performance by 2.4%, it still largely lags behind our MapVR. The improved performance from 'parallel segm' supports our motivation to enhance map vectorization via rasterization. However, we attribute its inferior performance compared to MapVR to the fact that, unlike in our MapVR, the fine-grained supervisory signal is not directly applied to the vectorized output.

## 5.4 Computational Overhead During Training

With the CUDA-accelerated differentiable rasterizer, our proposed MapVR only brings a marginal

Table 6: MapVR's computational overhead during the training stage. Results were obtained with 8x NVIDIA A100 GPUs under the same training setups.

| Method | Modality | Backbone | Training Time / Iter | GPU Memory Usage |
|---|---|---|---|---|
| MapTR [20] | C | Res-50 | 0.82 s | 14021 MB |
| MapTR [20] | C & L | Res-50 | 1.18 s | 28557 MB |
| MapTR [20] + MapVR (Ours) | C | Res-50 | 0.91 s | 14169 MB |
| MapTR [20] + MapVR (Ours) | C & L | Res-50 | 1.37 s | 28673 MB |

- *In modality, 'C' denotes multi-view camera input and 'C & L' denotes combined multi-view camera and LiDAR input.*

increase in memory footprint while maintaining training efficiency. Table 6 presents a detailed comparison between the training costs of our method, 'MapTR + MapVR', and its baseline, 'MapTR'.

## 5.5 Failure Case Analysis

While the proposed method greatly improves the quality of HD map vectorization, the numerical results suggest that the results are still far from perfect. We provide visualization of a few typical failure cases in Fig. 8.

From row 1, 2, and 4 in Fig. 8, it can be seen that occlusions, whether from vehicles, constructions, or a limited field of view, hamper perception in the bird's-eye-view. Such occlusions often result in inaccuracies in the predicted vectorized maps. Yet, since road structures typically follow regular patterns, current map vectorization techniques may benefit from integrating road structure priors, such as standard-definition maps (SDMap), to enhance their reasoning capabilities.

Row 5 in Fig. 8 shows that there is still room for improvement in nighttime driving.

Row 3 in Fig. 8 is caused by ambiguity in annotation, where it is unclear whether the middle crosswalk should connect to the adjacent ones or not.

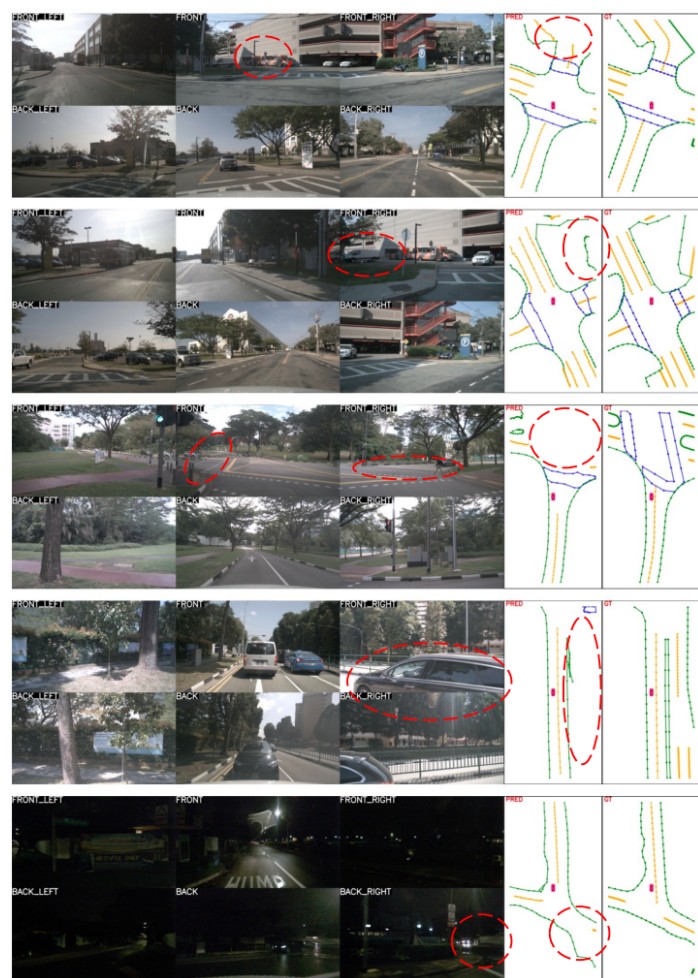

Figure 8: Visualization of failure cases produced by our method.

## 6 Conclusion

In this paper, we introduce a new perspective on map vectorization: rasterization, through which we can learn and evaluate map vectorization more precisely. We demonstrate that, while vectorized representation is compact and easy to use, it lacks representation capability, especially regarding fine-grained details; thus, it is necessary to incorporate rasterization as a complement in both learning and evaluation. We hope our perspective can serve as the cornerstone and spur further innovation in map vectorization, and can eventually lead to safe and reliable autonomous driving.

## Acknowledgement

The authors would like to thank the anonymous reviewers for their helpful suggestions.

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
