# Supplementary Materials

## Online Map Vectorization for Autonomous Driving: A Rasterization Perspective

**Gongjie Zhang**[†1]    **Jiahao Lin**[†1]    **Shuang Wu**[†1]    **Yilin Song**[1]    **Zhipeng Luo**[1,2]
**Yang Xue**[1]    **Shijian Lu**[2]    **Zuoguan Wang**[✉1]

[1]Black Sesame Technologies    [2]Nanyang Technological University, Singapore

† : equal contribution.    ✉ : corresponding author.    Project Page: https://github.com/ZhangGongjie/MapVR

## A    Technical Appendix

This technical appendix provides additional implementation details, more experimental results, and further discussions about our proposed MapVR as well as the new rasterization-based evaluation metric, which are omitted in the main paper due to space limitations.

### A.1    Implementation Details of MapVR

**Network Architectures.**    MapVR is a generic training paradigm that is directly applicable to any map vectorization model. To demonstrate the effectiveness of our proposed rendering-based training pipeline, we adopt the encoder-decoder-based network architecture from MapTR [4] as the base prediction model.

The base model takes surround-view images of the ego-vehicle as input. The model's encoder firstly extracts 2D image features for each camera view using a conventional convolution-based backbone. Following MapTR, we leverage GKT [1] to transform multi-view image features to a unified BEV space feature, which is used by the model's decoder to predict vectorized map elements. The decoder network consists of interleaved self-attention and cross-attention layers that progressively refine a set of queries. Specifically, the self-attention layers are implemented with Multi-Head Self-Attention (MHSA) [7] to enable the interaction among the queries, and the cross-attention layers are implemented with Deformable Attention [8] which attend to various locations in the BEV features [3]. Each query goes through a classification head for the class score prediction and a regression head for the vectorized point set prediction, respectively.

**Training Objectives.**    Given a set of predicted vectorized map elements and the set of ground truths, we adopt Hungarian Matching [2] to obtain the optimal assignment. The matching cost between each pair of prediction and ground truth instances is formulated as

$$\mathcal{L}^{(\text{Match})} = \lambda_1 \cdot \mathcal{L}_{\text{render}}^{(\text{Match})} + \lambda_2 \cdot \mathcal{L}_{\text{cls}}^{(\text{Match})} + \lambda_3 \cdot \mathcal{L}_{\text{reg}}^{(\text{Match})}. \tag{1}$$

The rendering cost $\mathcal{L}_{\text{render}}^{(\text{Match})}$ is implemented with a dice loss [6] between the softly rendered masks of the prediction and the ground truth. The classification cost $\mathcal{L}_{\text{cls}}^{(\text{Match})}$ is computed by applying the sigmoid function on the prediction's classification score of the particular class to which the matched ground-truth instance belongs. We use an L1-based regression loss $\mathcal{L}_{\text{reg}}^{(\text{Match})}$ with a small weight $\lambda_3$ to facilitate the matching process.

Once the optimal matching is obtained, we compute the final loss to supervise the training of the prediction model. The final loss for each prediction and its paired ground truth instance is defined as

$$\mathcal{L} = \lambda_1 \cdot \mathcal{L}_{\text{render}} + \lambda_2 \cdot \mathcal{L}_{\text{cls}} + \lambda_3 \cdot \mathcal{L}_{\text{dir}} + \lambda_4 \cdot \mathcal{L}_{\text{reg}}. \tag{2}$$

The rendering loss $\mathcal{L}_{\text{render}}$ and the regression loss $\mathcal{L}_{\text{reg}}$ are defined similarly as the costs in the matching process. The classification loss $\mathcal{L}_{\text{cls}}$ is implemented with a binary classification focal loss [5]. We further introduce the direction regularization loss $\mathcal{L}_{\text{dir}}$ on the predicted point set to regularize the regression output (See Eq. 4 in the main paper).

37th Conference on Neural Information Processing Systems (NeurIPS 2023).

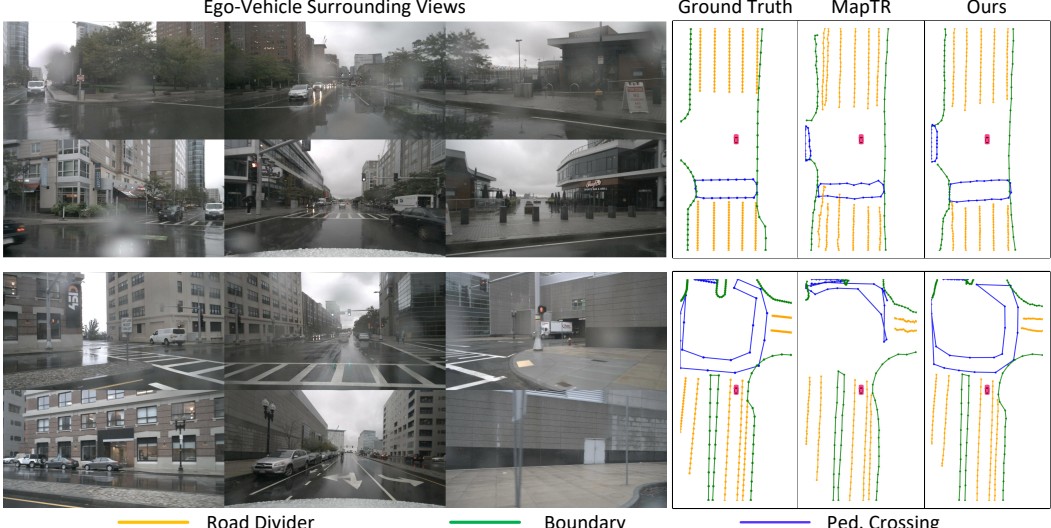

Figure 9: Additional visual comparison of online HD map vectorization results. Our proposed MapVR demonstrates a superior ability in constructing more accurate maps, particularly for complex map elements and intricate details.

## A.2 Implementation Details of $\text{AP}_{\text{raster}}$ (the Rasterization-Based Evaluation Metric)

Please visit our project page (https://github.com/ZhangGongjie/MapVR) for the implementation details of the $\text{AP}_{\text{raster}}$ metric.

Furthermore, we offer a standalone package of $\text{AP}_{\text{raster}}$ with simple instructions for usage, so that all researchers can adopt this metric for evaluation with ease. Please refer to https://github.com/jiahaoLjh/MapVectorizationEvalToolkit for the standalone implementation of the $\text{AP}_{\text{raster}}$ evaluation metric as well as the recommended hyper-parameter setups.

## A.3 Additional Experiment Results

As shown in Fig. 9, we provide further visual comparisons of HD map vectorization results. The results are consistent with our visualizations in the main paper: the proposed MapVR significantly enhances the model's capacity to perceive the finer details as well as those map elements with complex shapes. The results reaffirm the necessity of a rasterization perspective in map vectorization.

Fig. 10 presents more visualization of MapVR's HD map construction results. Our proposed method shows strong robustness across various scenes.

## A.4 Further Discussions

**Regarding Performance on Boundary Class.** In reference to the observed performance drop in $\text{AP}_{\text{Chamfer}}$ on the 'boundary' class in Table 1 of the main paper, we believe this is related to the curved nature of the boundary map elements and the lack of geometry awareness in $\text{AP}_{\text{Chamfer}}$. As shown in Fig. 10, these boundary map elements often embody a high number of curved or folded instances. As discussed in Section 3, the Chamfer-distance-based metric struggles to offer a fair evaluation for such scenarios. Therefore, we believe that this inherent limitation of the Chamfer distance primarily accounts for the performance drop in $\text{AP}_{\text{Chamfer}}$, and our proposed $\text{AP}_{\text{raster}}$ offers a more reasonable performance evaluation.

**Another Way to Understand MapVR.** MapVR is not just a training paradigm that bridges vectorized predictions and fine-grained HD map supervision. If viewed from an optimization perspective, MapVR is providing an extra dimension of supervision that complements regression-based losses. Specifically, the rasterization-based loss not only drives the prediction towards the ground truth, but also provides supervision in the direction that encourages better geometric alignment. This is verified by the experimental results in Fig. 7 in the main paper that when trained with only the

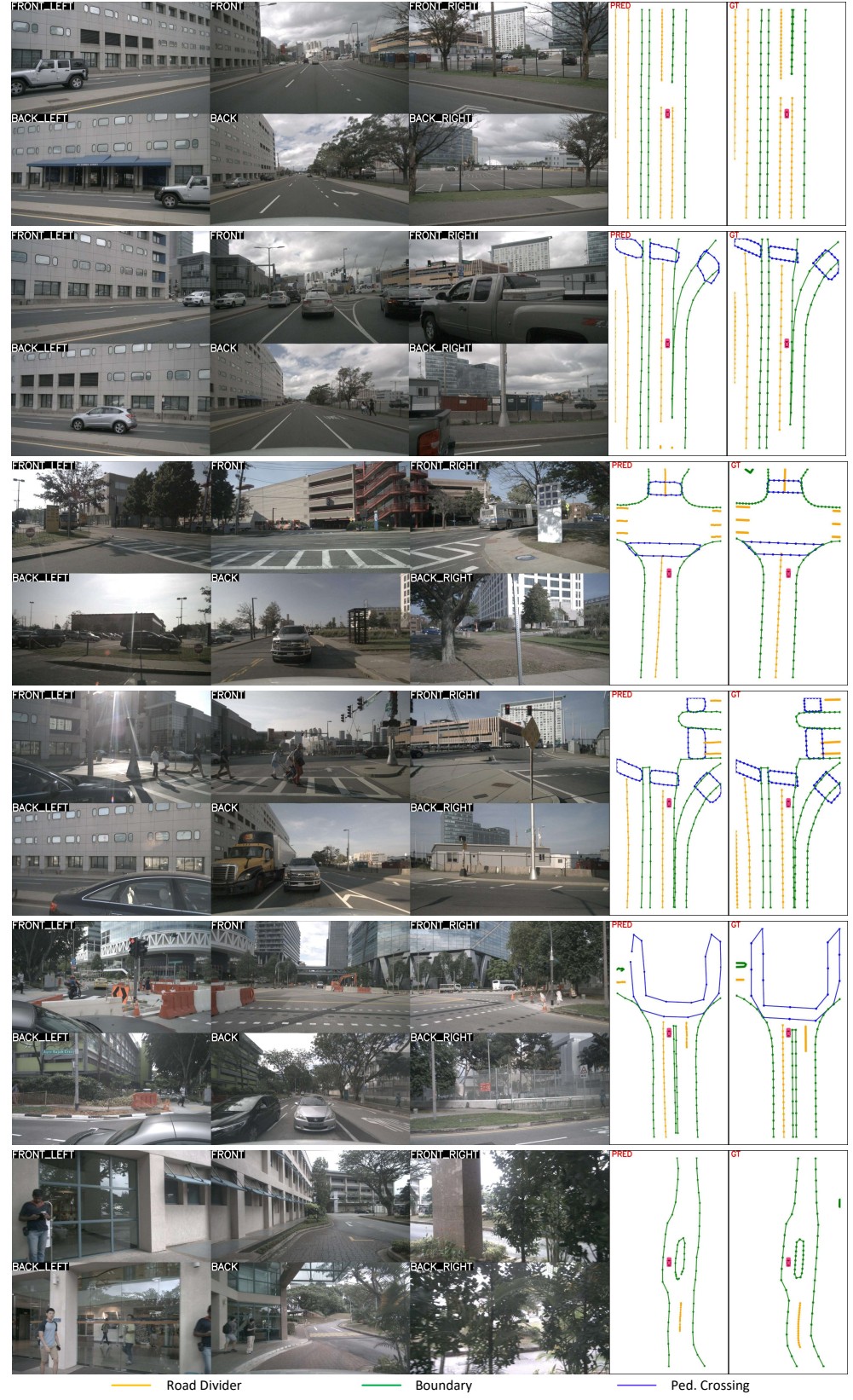

Figure 10: Visualization of the HD map construction results from our method (MapTR + MapVR).

regression-based loss, MapTR only performs well under the regression-based metric (*i.e.*, Chamfer distance) but much worse under the rendering-based metric since the geometric alignment is not enforced during training. It further demonstrates that our rendering-based evaluation metric is more comprehensive compared to the regression-based loss and is better suited for real-world autonomous driving scenarios.