# OpenReview forum: "Online Map Vectorization for Autonomous Driving: A Rasterization Perspective"
_NeurIPS.cc/2023/Conference — NeurIPS 2023 poster_

### Official Review · Reviewer_U7tP · 2023-07-06

**Soundness:** 4 excellent
**Presentation:** 3 good
**Contribution:** 3 good
**Rating:** 7
**Confidence:** 4

**Summary:**

This paper tackles the emerging task of vision-based online HD mapping. It has 2 main contributions: (1) AP_{raster}, a new metric that is shown to be better suited to evaluate methods in this field, and (2) MapVR, a plug-in rasterization and loss module to any existing vector-based online mapping system. Several intuitive examples demonstrate the value of the new metric. Furthermore, MapVR is extensively validated on three different datasets, where it maintains or improves the existing metrics, while significantly boosting the proposed AP_{raster} metric.

**Strengths:**

Both contributions are simple and well-motivated. The empirical evaluation is extensive, covering 4 task settings over 3 datasets. In particular, the design choices of MapVR are carefully ablated. In addition, the presentation and organization of the draft are good. Overall, this paper makes a crucial contribution to a field that is beginning to rapidly expand by identifying limitations in the current metrics employed in the literature and proposing an alternative.

**Weaknesses:**

No major weaknesses stood out to me. The presentation of the draft could be slightly improved (see “Questions” section). AP_{raster} may be sensitive to hyper-parameter choices, but I believe a supplementary experiment may be sufficient to demonstrate that the chosen hyper-parameters are reasonable, since they are well-motivated.

**Questions:**

1. What is the computational overhead of MapVR during training?
2. L30-32 and L96-106 have some text repetition. Would it be possible to rephrase the latter, focusing on the differences between different vector-based techniques (e.g. VectorMapNet and MapTR), instead of the motivation for MapVR?
3. In Tables 1, 2 and 3, would it be possible to highlight the “avg.” columns with a grey background, as in Table 5?
4. L243 “covering the most complex driving scenes in the real world” is a strong and unsubstantiated claim regarding this proprietary dataset. Please rephrase this.

Update:

Thank you for the thorough response. I appreciate the effort taken to answer each question. My concerns have been addressed in the rebuttal, and I am maintaining my initial positive rating.

**Limitations:**

Limitations are not discussed in main text - the main limitation I see is that AP_{raster} may be sensitive to hyper-parameter choices, which could be mentioned in the conclusion.

---

> ### Author Rebuttal · Authors · 2023-08-08
>
>
> &nbsp;
>
> We sincerely thank you for appreciating our work as well as for your thoughtful and constructive comments, which we believe will significantly enhance the presentation of our manuscript.
>
> &nbsp;
> ### Sensitivity of  $\text{AP}_\text{raster}$ to Hyper-Parameters
>
> We agree with the reviewer that $\text{AP}_\text{raster}$ relies on certain hyper-parameters, more specifically, the rasterization resolution (480x240) and the dilation pixel number (2 pixels). However, we think it does not constitute a severe limitation, but instead offers better evaluation flexibility.
>
> - _First_, the two hyper-parameters have clear practical meanings and are easy to set. In our nuScenes experiments, the bird's-eye-view (BEV) perception range is 60m x 30m. Setting the evaluation rasterization resolution at 480x240 offers a localization precision of 0.125m, satisfying the requirement of autonomous driving. A dilation of 2 pixels can make the evaluation criteria tolerant to a deviation of at most 4 pixels, which equals 0.5m. Users can easily adapt these hyper-parameters to their own needs.
>
> - _Second_, we conducted sensitivity analyses in the table below, which demonstrate that $\text{AP}_\text{raster}$ is robust with respect to the two hyper-parameters. Therefore, the performance improvements brought by MapVR under $\text{AP}\text{raster}$ are genuine and significant.
>
> |  | MapTR | MapTR+MapVR (Ours) |
> |---|:---:|:--:|
> | AP-raster (320x160, dilation=2) | 34.9 | 41.6 |
> | **AP-raster (480x240, dilation=2, default metric)** | 24.3 | 31.2 |
> | AP-raster (480x240, dilation=1) | 15.1 | 20.0 |
> | AP-raster (640x320, dilation=2) | 18.7 | 24.8 |
>
> - _Finally_, it's worth noting that most computer vision evaluation metrics (e.g., COCO's AP, nuScenes' NDS, etc.) have inherent hyper-parameters, and their choices always involve a trade-off between different considerations. The proposed $\text{AP}_\text{raster}$ is not an exception and we believe it does provide a valuable and practical tool for assessing the performance of HD map vectorization methods.
>
>
> &nbsp;
> ### Computational Overhead of MapVR During Training
>
> The extra differentiable rasterizer is implemented with CUDA to maintain training efficiency. The table below summarizes the training overhead.
>
> | Methods              | Modality | Backbone       | Training Time / Iter | GPU Memory Usage |
> |:---------------------|:--------:|:--------------:|:--------------------:|:----------------:|
> | MapTR                | C        | Res-50         | 0.82 s               | 14021 MB         |
> | MapTR                | C & L    | Res-50         | 1.18 s               | 28557 MB         |
> | MapTR + MapVR (Ours) | C        | Res-50         | 0.91 s               | 14169 MB         |
> | MapTR + MapVR (Ours) | C & L    | Res-50         | 1.37 s               | 28673 MB         |
>
>
> &nbsp;
> ### On the Presentation of the Manuscript
>
> We cannot be more grateful for your thoughtful suggestions. We will incorporate the following changes into the camera-ready version of the manuscript or future submissions.
>
> &nbsp;
>
> **To rephrase Line #96-106, focusing on the differences between map vectorization techniques.**
>
> Following your advice, we will revise Line #96-106 as below:
>
> >  Rasterization methods [37, 40, 12, 58, 19, 59, 36, 50] generate HD maps via semantic segmentation in BEV, which have good sensitivity to details. However, the lack of vital instance-level information and lane topology limits the utility of rasterized maps in downstream tasks like navigation and planning. On the other hand, map vectorization addresses this limitation by producing vectorized map elements. HDMapNet [16] and SuperFusion [6] employ post-processing to group pixels from rasterized maps into vectorized elements. ~~Moreover, the latest approaches – VectorMapNet [26] and MapTR [20], directly predict map elements as vectorized point sets with neural networks, attaining superior performance.~~ _Moreover, VectorMapNet [26] proposes to directly predict map elements as vectorized point sets in an auto-regressive manner, achieving superior performance. And MapTR [20] - the current state of the art, further proposes a unified permutation-equivalent modeling approach to model the HD map elements, achieving superior accuracy. Furthermore, MapTR [20] achieves real-time efficiency with its one-stage and parallel framework._ However, _despite the recent progresses,_ vectorized maps still often exhibit minor deviations that can be critical in autonomous driving, where safety is of utmost importance. ~~We believe integrating HD rasterization into vectorization can improve precision while retaining vectorized representation.~~
>
>
> &nbsp;
>
> **To highlight the “avg.” columns in Table 1-3 with a grey background.**
>
> Sure, we will do so in the updated version.
>
>
> &nbsp;
>
> **To rephrase Line #243.**
>
> The suggested sentence will be rephrased into:
> > covering ~~the most~~ _very_ complex driving scenes in the real world.
>
>
> &nbsp;
> ### Additional Discussions on Limitations
>
> Due to the space limitation, we did not discuss the limitations in the submitted manuscript. In the camera-ready version where one additional content page is allowed, we will incorporate a discussion on the potential limitations.
>
> **_Please refer to `our global response to all reviewers` for additional discussions on potential limitations._**
>
>
> &nbsp;
>
> &nbsp;
>
> We hope that our response has addressed your concern. And we welcome further dialogue to discuss any concerns and clear up any doubts that may still exist.
>
> &nbsp;

---

> > ### Comment · Reviewer_U7tP · 2023-08-19
> >
> > Thank you for the thorough response. I appreciate the effort taken to answer each question. My concerns have been addressed in the rebuttal, and I am maintaining my initial positive rating.

---

> > > ### Author Response · Authors · 2023-08-19
> > > **Thank you!**
> > >
> > > Thank you for spending time to read our rebuttal. We are glad that your concerns have been addressed.
> > >
> > >
> > > Once again, we sincerely thank you for recognizing the merits of our work!

---

### Official Review · Reviewer_pGkQ · 2023-07-06

**Soundness:** 3 good
**Presentation:** 2 fair
**Contribution:** 2 fair
**Rating:** 4
**Confidence:** 5

**Summary:**

This work targets online map creation for autonomous driving.
The authors claim vector-based approaches exhibit artifacts due to lack of geometric supervision in current loss functions.
The main idea is to add a differentiable rasterization layer to any model that predicts vectors and add an additional segmentation loss (dice loss) during training.
Their method does not require any additional model parameters, and can be added to any existing vector-based model.
They perform experiments on the nuScenes dataset using camera only, and camera + lidar,
and show their additional supervision can improve performance of the current state-of-the-art vector-based model.

**Strengths:**

* Simple idea, easy to execute and shows modest improvements for MapTR with camera+lidar.
* The proposed method can be applied to any vector-based architecture.
* Technical section described clearly, notation looks sound.
* Convincing analysis and visual evidence for the need of the rasterization-based metric (Figure 4.)

**Weaknesses:**

* I am borderline on if the current contribution is significant enough for a paper. The results are sound, particularly for AP-raster since the baselines do not optimize for this. However, the story and writing needs more focus on the AP-raster metric if that is the real contribution (i.e. should all future map models focus on this particular metric?).
* The main ideas presented are architecture agnostic but experiments are only done using MapTR.
* I found the direction regularization loss to be unrelated to the main idea of rasterization. How does the baseline (MapTR) perform with this extra loss, or how does MapVR perform without this? A majority of the main results in Table 1 are very close - this is my primary factor for my current rating.
* I was hoping ablations included evaluation using both metrics (AP-raster+AP-chamfer).

**Questions:**

* Table 1: It would be nice to have AP-raster numbers for HDMapNet and VectorMapNet. In the current draft these two methods do not provide much information as of now.
* Should all future vector models be evaluated and assessed with AP-chamfer and AP-raster metrics? Is the proposed AP-raster superior?
* Figure 2: the authors argue that equidistant parameterization causes inaccuracy in modeling map primitives. Couldn't the same be said for resolution of rasterization? Why not simply increase the number of points?

**Limitations:**

No limitations/societal impact section provided.

---

> ### Author Rebuttal · Authors · 2023-08-09
>
>
> &nbsp;
>
> Thank you for taking the time to review our manuscript. Your comments have provided us with valuable insights to improve our work. We have carefully considered each of your points, and we hope that our responses below could address your concerns.
>
> &nbsp;
> ### Significance of $\text{AP}_\text{raster}$
>
> Short answer: **We believe $\text{AP}_\text{raster}$ is of great significance to the autonomous driving community, and we highly recommend future works to use** $\text{AP}_\text{raster}$ **for map vectorization evaluation.**
>
> Actually, in our team, we already transitioned to $\text{AP}_\text{raster}$. And we believe it is also valuable for the emerging community, as it guides subsequent works toward the actual needs of autonomous driving.
>
> To highlight its significance, we have dedicated the whole of Section 3 to introducing this evaluation metric. And we will amend the narratives to make our points clearer.
>
> We further explain the significance of $\text{AP}_\text{raster}$ as follows.
> - First, as described in Section 3.1, the existing AP_chamfer metric is not scale-invariant and overlooks shape and geometrical details. Fig. 4 illustrates these issues with a few practical examples, where AP_chamfer fails to reasonably judge the prediction quality while AP_raster does clearly a better job.
> - Second, from the visualizations in Fig. 6, it can be observed that our proposed rasterization-based approach yields substantially better results, although it does not outperform MapTR by a large margin in terms of AP_chamfer. This also indicates that AP_chamfer falls short of providing a comprehensive and precise evaluation as compared to AP_raster.
> - Third, while the judgment of significance can be subjective, `Reviewers YAhf and U7tP` strongly support us on the significance of $\text{AP}\text{raster}$. Their comments reflect the value of this new metric to the community to some extent. Nevertheless, we fully appreciate your concern and welcome further conversations.
>
>
> &nbsp;
> ### Compatibility of MapVR
>
> We agree we should have integrated MapVR with more baselines. However, as map vectorization is an emerging field, existing studies are limited.
>
> By the time of submission (May 2023), VectorMapNet and MapTR are the only two peer-reviewed works that directly use networks to predict the vectorized map elements. Since MapTR outperforms VectorMapNet significantly in both accuracy and efficiency, we selected MapTR as the baseline.
>
> Besides, as our work does not involve any assumption or modification in network architecture, it should be generalizable to other future map vectorization solutions.
>
>
> &nbsp;
> ### Why Not Increase the Number of Equidistant Points to Avoid Parameterization Error?
>
> The attempt was made in MapTR, but resulted in a decline in performance. You can find more information about this in Table 5 of MapTR's appendix [20]. This issue has also been mentioned in the footnote of Page 2 in our manuscript.
>
>
> &nbsp;
> ### Regarding Direction Regularization Loss
>
> We wish to clarify that **MapTR's official results were obtained with a similar direction loss**. Therefore the comparisons in Tables 1-3 are fair.
>
> MapTR's direction loss is computed between predicted points and ground-truth equidistant points; while our MapVR's direction loss is self-regularized along predicted adjacent segments. With the supervision via differentiable rasterization, MapVR largely diminishes the need for equidistant predicted points, which was a constraint in MapTR. As a result, our designed direction loss aids in allocating more points in areas with greater curvature and fewer points in straight-line regions, ultimately enhancing the precision.
>
> In conclusion, the direction regularization loss is not unrelated to the rasterization idea, but serves as an important component that works in conjunction with it for better map vectorization.
>
> The table below presents extra experiments regarding the direction loss.
>
> |  |MapTR   (AP-chamfer/AP-raster)|MapTR+MapVR(Ours)  (AP-chamfer/AP-raster)|
> |--|:--:|:--:|
> |w/o direction loss|48.2 / 23.7| 48.5 / 29.5 |
> |w/ MapTR's direction loss|50.3 / 24.3|---- / ----|
> |w/ our direction loss|---- / ----| 51.2 / 31.2 |
>
>
> &nbsp;
> ### Regarding Blanks in Table 1
>
> Neither HDMapNet nor VectorMapNet provides source codes for the "C & L" modality. Besides, HDMapNet does not provide any checkpoint for evaluation. Therefore, we did not provide their performance under $\text{AP}_\text{raster}$.
>
> Fortunately, VectorMapNet provides codes and checkpoint for the camera-only modality. The table below shows the additional results (highlighted in bold), which we will incorporate into future versions.
>
> |Method|Modality|Backbone|#Epochs|AP_chamfer_avg|AP_raster_avg|FPS|
> |--|:--:|:--:|:--:|:--:|:--:|:--:|
> |VectorMapNet|C|Res-50|110|40.9|**15.0**|2.9|
> |MapTR|C| Res-50 |110|58.7|35.0|18.4|
> |MapTR+MapVR (Ours)|C|Res-50|110|58.8|38.5|18.4|
>
> It can be seen that: for methods that are inferior under $\text{AP}\text{chamfer}$, the gap under the stricter metric $\text{AP}\text{raster}$ will be even larger.
>
>
> &nbsp;
> ### Ablations Under Both Metrics
>
> (a) Rasterization resolution
>
> | resolution | AP-chamfer/AP-raster |
> |--|:--:|
> |X | 50.3 / 24.3 |
> |64x32 | 45.1 / 21.5 |
> |180x90 | 50.6 / 30.4 |
> |256x128 | 51.2 / 31.2 |
> |320x160 | 50.9 / 30.9 |
>
> &nbsp;
>
> (b) Line rasterization softness
>
> | $\tau$ | [Divider] AP-chamfer/AP-raster |
> |--|:--:|
> |0.5|48.0 / 29.2|
> |2.0|54.4 / 33.1|
> |6.0|52.8 / 31.4|
>
> &nbsp;
>
> (c) Direction loss.
>
> _Please refer to the response above ('Regarding Direction Regularization Loss')._
>
> &nbsp;
>
> (d) Geometry-aware rasterization
>
> | | [Ped_crossing] AP-chamfer/AP-raster |
> |--|:--:|
> |all as lines|34.9 / 21.8|
> |lines and polygons|47.7 / 37.5|
>
> &nbsp;
>
> (e) MapVR vs. parallel segm
>
> | |AP-chamfer/AP-raster|
> |--|:--:|
> |MapVR|51.2 / 31.2|
> |parallel segm|48.1 / 26.7|
>
> &nbsp;
>
> &nbsp;
>
> We sincerely hope the above address your concerns. Please don't hesitate to continue the discussion if you have further queries.
> &nbsp;

---

> > ### Comment · Reviewer_pGkQ · 2023-08-21
> >
> > Thank you for clarifying some of these important details and running additional experiments during the short rebuttal period.
> > I would highly recommend adding these into the revision!
> >
> > It's interesting to see the field go to vectorized representation then back to using the raster metric but I am more convinced now on the usefulness of this work.
> >
> > I'll raise my score post-rebuttal

---

> > > ### Author Response · Authors · 2023-08-21
> > > **Thank you!**
> > >
> > > Thank you for taking time to read our response! We are so glad and grateful to have your acknowledgement!
> > >
> > > We will surely add them into our final version. And we are very grateful for your constructive comments that help us improve the manuscript.

---

### Official Review · Reviewer_YAhf · 2023-07-06

**Soundness:** 4 excellent
**Presentation:** 4 excellent
**Contribution:** 4 excellent
**Rating:** 8
**Confidence:** 4

**Summary:**

Online HD mapping is essential for autonomous driving because maps provide detailed and precise environmental information for perception and planning. Existing mapping literatures have limitations both in methods and metrics. To address them, the authors a new method called MapVR, integrating the philosophy of rasterization into map vectorization and a new metric, a rasterization-based evaluation metric. Experiments are conducted on four existing datasets (nuScenes Map (basic), nuScenes Map (extended), Argoverse2, and 6V-mini-v0.4) and show that incorporating rasterization into map vectorization greatly enhances performance with no extra computational cost during inference.

**Strengths:**

1. Writing: Overall the paper is well-written and is easy to understand the goal of the paper.

2. Motivation: Particularly, the authors provide a very introduction that explains the limitations of existing mapping algorithms and metrics. The work is highly valuable to the field of HD mapping.

3. Proposed method: To address the limitations of existing mapping algorithms, the authors propose to combine the two camps (i.e., map rasterization and map vectorization) in a unified framework. In particular,  the proposed MapVR applies differentiable rasterization, inspired by recent advances in graphics and vision, to the map vectorization task to bridge vectorized outputs and rasterized HD maps. The proposed strategy enables more refined and comprehensive supervision and yields predictions with improved precision. The authors offer a new perspective on HD mapping. Moreover, the work could motivate relevant fields in autonomous vehicle research.

4. Metric: the authors deliver a clear explanation of the proposed metric in Figures 3 and 4. Particularly, Figure 4 can immediately spot the effect of two different metrics. Importantly, the proposed metric can capture fine-grained differences for quality evaluation.

5. Experimental results: the authors conduct experiments on multiple datasets, i.e., nuScenes Map (basic), nuScenes Map (extended), Argoverse2, and 6V-mini-v0.4. In Table 1-4, MapVR clearly demonstrates its superiority over existing baselines (i.e., HDMapNet, VectorMapNet, MapTR). The tables provide a clear benchmarking protocol for the community. It is worth noting that MapVR can enhance map vectorization without adding any extra computational cost during inference.

6. Ablation study: the reviewer particularly finds the “1. **Why Not Introduce HD Supervisory Signals from an Auxiliary Segmentation Task?**” important. It shows that a simple auxiliary task can benefit rasterization prediction. The results are valuable for future research.

**Weaknesses:**

Overall, the paper is a really good shape. However, the reviewer has the following comments.

1. Differentiable rasterization: In sec. 4.2, the authors discussed two rendered masks for line-shaped and polygon-shaped elements. Can the two rendered masks cover all elements in HD maps? On a relevant note, do the authors identify failure cases due to the conversion?

2. Rasterization resolution: what is the implication of having 256x128 the best performance? Is it related to the HD map size? Do we have a systematic way to identify the proper rasterization resolution?

3. While the proposed method achieves the best quality, the numerical results are still far from perfect. The reviewer strongly suggests the authors conduct a thorough failure case analysis that would shed light on future research.

**Questions:**

Please see the weakness section.

**Limitations:**

No. The authors do not provide adequate discussions on the limitations.

---

> ### Author Rebuttal · Authors · 2023-08-09
>
> &nbsp;
>
> We are immensely grateful for the comprehensive and positive acknowledgment received. And we deeply appreciate your insightful comments and suggestions. Below we would like to respond to your queries and address your concerns point-by-point.
>
> &nbsp;
>
> &nbsp;
> ### Discussions on Failure Cases and Limitations
>
> We greatly appreciate your constructive feedback. If our work is accepted for NeurIPS 2023, we will incorporate discussions on failure cases and limitations within the one additional page allowed in the camera-ready version.
>
> **_Please refer to `our global response to all reviewers` for (i) additional visualization and analysis on failure cases and (ii) additional discussions on limitations._**
>
> &nbsp;
> ### Regarding Line-Shaped and Polygon-Shaped Differentiable Rasterization
>
> Our empirical findings indicate that the two proposed rasterization strategies can effectively cover all elements in HD maps.
>
> To the best of our knowledge, the majority of HD map elements can be abstracted into either one of three primal shapes: points, lines, and polygons. As thoroughly explained in our manuscript, the most common HD map elements, such as lanes, curbsides, stoplines, crosswalks, intersections, and parking areas, can be categorized as lines and polygons, which can be well covered by the two designed rasterization strategies.
> Other elements like traffic lights, traffic signs, and cones are ideally abstracted as points within HD maps. These elements can be interpreted as small, circular-shaped polygons, thus integrating them seamlessly into our designed rasterization strategies. This strategy allows us to integrate our proposed method in an 'all-in-one' vectorized map perception framework without any complex network design.
>
> It should be noted that the differentiable rasterization procedure is accomplished intuitively via Eq. 2 and Eq. 3 in our manuscript, rather than learned parametrically. We did not notice any error caused by the differentiable rasterizer.
>
>
>
> &nbsp;
> ### Regarding Rasterization Resolution
>
> In our understanding, this hyper-parameter is related to the perception range as well as the perception precision.
>
> Taking our nuScenes experiments as an example. The perception range is ±30m along the y-axis (vertically) and ±15m along the x-axis (horizontally), yielding a bird's-eye-view (BEV) perception range of 60m x 30m. With a rasterization resolution of 256x128 pixels, each rasterized pixel corresponds to a real-world area of 0.23m x 0.23m. This precision is generally considered sufficient for accurate fine-grained localization supervision.
>
> To determine the appropriate rasterization resolution, it is crucial to proportionally align the rasterization resolution with the BEV perception range. For example, if we keep the per-pixel size unchanged, a perception range of 120m x 45m would correspond to a rasterization resolution of 480x180. In addition, fine-tuning the rasterization resolution within a small neighboring range should help for better performance.
>
> It is also noteworthy that, as per Table 5a in our manuscript, the performance is robust against a certain range of rasterization resolutions (180x90 ~ 320x160).
>
> &nbsp;
>
> &nbsp;
>
> We hope the above response addresses your concerns. Once again, we thank you for your insightful comments and strong recognition of our work, and we are more than happy to have further discussions with you.
>
> &nbsp;

---

> > ### Comment · Reviewer_YAhf · 2023-08-18
> > **Response to the rebuttal. Thanks!**
> >
> > Dear Authors,
> >
> > Thanks for the detailed response! Most of the questions are addressed! I have a follow-up question and suggestion:
> >
> > 1. I am particularly interested in failure example #3, shown in the rebuttal PDF. I would like to learn from the authors the reason behind it because it is not a challenging case.
> >
> > 2. Regarding Line-Shaped and Polygon-Shaped Differentiable Rasterization: Thanks for the detailed response. I felt it would be a good summary if the authors can provide several examples of lanes, curbsides, stoplines, crosswalks, intersections, parking areas, traffic lights, traffic signs, and cones in the appendix in the final version to showcase the proposed method can reconstruct these elements well.

---

> > > ### Author Response · Authors · 2023-08-19
> > > **Authors' Response to Reviewer's Follow-Up Inquiry and Suggestion**
> > >
> > > Thanks for reading our rebuttal. We are glad that most of your concerns have been addressed.
> > >
> > > As per your follow-up inquiry and suggestion, we would like to respond below.
> > >
> > >
> > >
> > > - There are two reasons that we attribute to _failure case #3_.
> > >
> > >     - _First_, there exists ambiguity in the connectivity of crosswalk ground truth. It is unknown whether the left vertical crosswalk and the right vertical crosswalk should be connected to the horizontal crosswalk or not. In _failure case #3_, the left crosswalk is separate and the right crosswalk is linked to the horizontal one, while there is no visual clue to distinguish these two patterns. The ambiguity causes failure in detecting the left and the right crosswalks.
> > >
> > >     - _Second_, intersections with a yellow box on the road face are underrepresented in the nuScenes dataset. This causes unsatisfactory results in _failure case #3_, as the network rarely sees such cases during training.
> > >
> > >
> > >
> > > - Thank you for your advice. We will surely update the final version accordingly.
> > >
> > > &nbsp;
> > >
> > > Once again, thank you for your valuable advice as well as your strong recognition of our work.

---

> > > > ### Comment · Reviewer_YAhf · 2023-08-19
> > > > **Thank you for the prompt response!**
> > > >
> > > > Dear authors,
> > > > Thanks for the prompt response! Please include the reasons for the failure in the final version.
> > > > Thanks again for sharing your work, and I learn a lot for it!

---

> > > > > ### Author Response · Authors · 2023-08-19
> > > > > **Thank you!**
> > > > >
> > > > > Will do.
> > > > >
> > > > > We really enjoy our discussion! Thanks for the helpful suggestions!

---

### Official Review · Reviewer_yTjh · 2023-07-13

**Soundness:** 3 good
**Presentation:** 3 good
**Contribution:** 2 fair
**Rating:** 4
**Confidence:** 4

**Summary:**

--

**Strengths:**

--

**Weaknesses:**

--

**Questions:**

--

**Limitations:**

--

---

### Official Review · Reviewer_ddVN · 2023-07-27

**Soundness:** 3 good
**Presentation:** 3 good
**Contribution:** 2 fair
**Rating:** 5
**Confidence:** 2

**Summary:**

The authors explore and analyze the existing HD map framework and its evaluation metrics, and propose to adopt rasterization on top of the existing vectorization-based HD map for more precision. Also, they propose a rasterization-based evaluation metric rather than the existing chamfer-distance one. The experimental results on a number of datasets are better compared to the existing methods.

**Strengths:**

1. The task of HD map is very important in the 3D autonomous driving community. Interestingly, the authors propose to adopt rasterization on top of the existing vectorization-based HD map for more precision.

2. The paper is easy to follow.

3. The authors conduct the experiments on differnt outdoor datasets, including widely-used nuScenes, Argoverse2 and their own 6V-mini-v0.4.

**Weaknesses:**

1. Computation/memory footprint comparison. The authors didn't make a comparison of their work in terms of memory/speed with the existing 3D HD map methods. The time consumption might be not trivial since the memory/time might be heavy for the customized differentiable rasterizer from my point.

**Questions:**

Please refer to the questions that I describe in the Weakness part. I would also consider the rebuttal and other reviews.

**Limitations:**

NA.

---

> ### Author Rebuttal · Authors · 2023-08-08
>
>
> &nbsp;
>
> Thank you for acknowledging the significance, presentation, and thorough evaluation of our work. We also appreciate your kind suggestion on computation/memory footprint comparison.
>
> We hope the information provided below will help address your concerns. And we will include them in our final version.
>
> &nbsp;
>
> &nbsp;
>
> ## Computation/Memory Footprint (Evaluation Metric)
>
> Our empirical tests indicate that our rasterization-based evaluation metric $AP_\text{raster}$, though more complex, runs at similar speeds to the Chamfer-distance-based metric $AP_\text{chamfer}$. Specifically, our metric takes about 3 minutes to evaluate on the nuScenes Map validation set, while the Chamfer-distance-based metric takes around 2 minutes. Notably, our proposed evaluation metric can be further accelerated with multi-threading.
>
> Both evaluation metrics are performed on CPU sequentially, therefore their memory consumption is negligible.
>
> &nbsp;
>
> &nbsp;
>
> ## Computation/Memory Footprint (Neural-Network Model)
>
>
> ### Training Stage
>
> We concur with the reviewer that the runtime/memory consumption will be expensive if the customized differentiable rasterizer is implemented in pure PyTorch/TensorFlow. Given this, we implement the differentiable rasterizer (both forward and backward propagation) with CUDA. Please refer to our code implementation in the supplementary materials for details. Upon acceptance, we intend to release all our code implementations. This will allow our work to serve as a foundation for future research and development.
>
> With the CUDA-accelerated differentiable rasterizer, our proposed MapVR ensures a marginal increase in memory footprint while maintaining computational efficiency during the training process. The following table presents a detailed comparison between the training costs of our method, `MapTR+MapVR`, and its baseline, `MapTR`. Experiments were conducted with 8x NVIDIA A100 GPUs under the training setups specified in our manuscript.
>
> &nbsp;
>
> | Methods              | Modality | Backbone       | Training Time / Iter | GPU Memory Usage |
> |:---------------------|:--------:|:--------------:|:--------------------:|:----------------:|
> | MapTR                | C        | Res-50         | 0.82 s               | 14021 MB         |
> | MapTR                | C & L    | Res-50         | 1.18 s               | 28557 MB         |
> | MapTR + MapVR (Ours) | C        | Res-50         | 0.91 s               | 14169 MB         |
> | MapTR + MapVR (Ours) | C & L    | Res-50         | 1.37 s               | 28673 MB         |
>
> &nbsp;
> ### Inference Stage
>
> The differentiable rasterization is not needed anymore once the training is complete. Hence, our method can improve map vectorization with no additional computation or memory usage during inference.
>
> In Table 1 and Table 2 of our manuscript, we have compared the inference speed of different methods for map vectorization. The table below is an updated version of Table 1, which provides more details on inference time, memory consumption, etc. Results are obtained with 1x NVIDIA 3090 GPU.
>
> &nbsp;
>
> | Method             | Modality | Backbone | #Epochs | AP_chamfer_avg | AP_raster_avg | FPS  | GPU Memory |
> |:-------------------|:--------:|:--------:|:-------:|:--------------:|:-------------:|:----:|:----------:|
> | HDMapNet           | C        | Effi-B0  | 30      | 23.0           | -             | 0.8  | 3264 MB    |
> | HDMapNet           | C & L    | Effi-B0  | 30      | 31.0           | -             | 0.5  | -          |
> | VectorMapNet       | C        | Res-50   | 110     | 40.9           | 15.0          | 2.9  | 3232 MB    |
> | VectorMapNet       | C & L    | Res-50   | 110     | 45.2           | -             | -    | -          |
> | MapTR              | C        | Res-50   | 24      | 50.3           | 24.3          | 18.4 | 3115 MB    |
> | MapTR              | C        | Res-50   | 110     | 58.7           | 35.0          | 18.4 | 3115 MB    |
> | MapTR              | C & L    | Res-50   | 24      | 62.7           | 45.2          | 7.2  | 16760 MB   |
> | MapTR+MapVR  (Ours)| C        | Res-50   | 24      | 51.2           | 31.2          | 18.4 | 3115 MB    |
> | MapTR+MapVR  (Ours)| C        | Res-50   | 110     | 58.8           | 38.5          | 18.4 | 3115 MB    |
> | MapTR+MapVR  (Ours)| C & L    | Res-50   | 24      | 63.5           | 51.1          | 7.2  | 16760 MB   |
>
> Note: Entries marked with '-' indicate that official results are not reported, and codes or model checkpoints are unavailable.
>
>
> &nbsp;
>
> &nbsp;
>
> We hope that our response has addressed your concern. And we welcome further dialogue to discuss any concerns and clear up any doubts that may still exist.
>
> &nbsp;

---

> > ### Comment · Reviewer_ddVN · 2023-08-21
> > **Authors have addressed most of my concerns.**
> >
> > Thanks for the answers and clarification in the rebuttal, which covered most of my concerns.

---

> > > ### Author Response · Authors · 2023-08-21
> > > **Thank you!**
> > >
> > > Thank you so much for your reply. We are so glad to hear that!
> > >
> > > We will add these results into our final version. And we sincerely appreciate your effort and your advice during the review phase.

---

### Author Rebuttal · Authors · 2023-08-09

&nbsp;

We are incredibly grateful for all the helpful comments received.

Here, we provide some additional contents that cannot fit into the nine-page manuscript. These additional contents will be included in the camera-ready version, in which one additional page of content is allowed.

&nbsp;

&nbsp;

## Potential Limitations

We found the potential limitations of our work primarily regarding the reliance on certain hyper-parameters.

&nbsp;

### Sensitivity of $\text{AP}_\text{raster}$ to Hyper-Parameters

Our designed $\text{AP}_\text{raster}$ relies on certain hyper-parameters, more specifically, the rasterization resolution (480x240) and the dilation pixel number (2 pixels). However, we think it does not constitute a severe limitation, but instead offers better evaluation flexibility.

- _First_, the two hyper-parameters have clear practical meanings and are easy to set. In our nuScenes experiments, the bird's-eye-view (BEV) perception range is 60m x 30m. Setting the evaluation rasterization resolution at 480x240 offers a localization precision of 0.125m, satisfying the requirement of autonomous driving. A dilation of 2 pixels can make the evaluation criteria tolerant to a deviation of at most 4 pixels, which equals 0.5m. Users can easily adapt these hyper-parameters to their own needs.

- _Second_, we conducted sensitivity analyses in the table below, which demonstrate that $\text{AP}_\text{raster}$ is robust with respect to the two hyper-parameters. Therefore, the performance improvements brought by MapVR under $\text{AP}\text{raster}$ are genuine and significant.

|  | MapTR | MapTR+MapVR (Ours) |
|---|:---:|:--:|
| AP-raster (320x160, dilation=2) | 34.9 | 41.6 |
| **AP-raster (480x240, dilation=2, default metric)** | 24.3 | 31.2 |
| AP-raster (480x240, dilation=1) | 15.1 | 20.0 |
| AP-raster (640x320, dilation=2) | 18.7 | 24.8 |

- _Finally_, it's worth noting that most computer vision evaluation metrics (e.g., COCO's AP, nuScenes' NDS, $\text{AP}\text{chamfer}$, etc.) have inherent hyper-parameters, and their choices always involve a trade-off between different considerations. The proposed $\text{AP}_\text{raster}$ is not an exception and we believe it does provide a valuable and practical tool for assessing the performance of HD map vectorization methods.


&nbsp;

### Sensitivity of MapVR to Hyper-Parameters

MapVR also introduces hyper-parameters that could affect performance. The major concern is on the rasterization resolution.

Similarly, this hyper-parameter is most related to the perception range as well as the perception precision.

Taking our nuScenes experiments as an example. The perception range is ±30m along the y-axis (vertically) and ±15m along the x-axis (horizontally), yielding a bird's-eye-view (BEV) perception range of 60m x 30m. With a rasterization resolution of 256x128 pixels, each rasterized pixel corresponds to a real-world area of 0.23m x 0.23m. This precision is generally considered sufficient for accurate fine-grained localization supervision.

It should be quite simple to customize the hyper-parameters under a particular setup. To determine the appropriate rasterization resolution, it is crucial to proportionally align the rasterization resolution with the BEV perception range. For example, if we keep the per-pixel size unchanged, a perception range of 120m x 45m would correspond to a rasterization resolution of 480x180. In addition, fine-tuning the rasterization resolution within a small neighboring range should help for better performance.

It is also noteworthy that, as per Table 5a in our manuscript, the performance is robust against a certain range of rasterization resolutions (180x90 ~ 320x160).




&nbsp;

&nbsp;

## Failure Case Analysis

Based on Fig. 6 in our manuscript as well as the visualization results in the appendix, it is evident that our method can perceive pretty accurately when the road structure is highly regularized and when there is no occlusion.

**In the `pdf` file attached in this 'global rebuttal' message, we present a few failure cases for analysis.**

These cases reveal challenges predominantly at complex road intersections. Here, occlusions, whether from vehicles, constructions, or a limited field of view, hamper our system's perception in the bird's-eye-biew (BEV). Such occlusions often result in inaccuracies in the predicted vectorized maps. Yet, since road structures typically follow regular patterns, there's an opportunity for improvement. Current map vectorization techniques may benefit from integrating road structure priors and enhancing their reasoning capabilities. Future research could explore merging perception frameworks with road prior information or leveraging the knowledge from the standard definition map (SDMap) to address these challenges. Moreover, enhancing map perception during nighttime remains an exciting direction for upcoming works.







&nbsp;

---

### Decision · Program_Chairs · 2023-09-21

**Decision:**

Accept (poster)

**Comment:**

Existing online mapping works have limitations both in methods and metrics. To address them, this paper proposes a new method called MapVR, integrating the idea of rasterization into map vectorization and a new rasterization-based evaluation metric. It receives one borderline accept, two borderline rejects, one strong accept and one accept. Reviewer yTjh fails to provide detailed comments thus the rating is ignored by the AC. Authors provided detailed response and failure case analysis during the rebuttal. AC recommends acceptance for this paper.